# SARS-CoV-2 Nucleocapsid Protein Induces Tau Pathological Changes That Can Be Counteracted by SUMO2

**DOI:** 10.3390/ijms25137169

**Published:** 2024-06-28

**Authors:** Franca Orsini, Marco Bosica, Annacarla Martucci, Massimiliano De Paola, Davide Comolli, Rosaria Pascente, Gianluigi Forloni, Paul E. Fraser, Ottavio Arancio, Luana Fioriti

**Affiliations:** 1Department of Neuroscience, Istituto di Ricerche Farmacologiche Mario Negri IRCCS, 20156 Milano, MI, Italy; franca.orsini@marionegri.it (F.O.); marco.bosica@marionegri.it (M.B.); annacarla.martucci@marionegri.it (A.M.); massimiliano.depaola@marionegri.it (M.D.P.); davide.comolli@marionegri.it (D.C.); rosaria.pascente@marionegri.it (R.P.); gianluigi.forloni@marionegri.it (G.F.); 2Tanz Centre for Research in Neurodegenerative Diseases, University of Toronto, Toronto, ON M5T 2S8, Canada; paul.fraser@utoronto.ca; 3Department of Pathology and Cell Biology, Taub Institute for Research of Alzheimer’s Disease and the Aging Brain, Columbia University, New York, NY 10032, USA; oa1@columbia.edu

**Keywords:** SARS-CoV-2, nucleocapsid protein, tau, stress granules, SUMO, synapses, memory

## Abstract

Neurologic manifestations are an immediate consequence of SARS-CoV-2 infection, the etiologic agent of COVID-19, which, however, may also trigger long-term neurological effects. Notably, COVID-19 patients with neurological symptoms show elevated levels of biomarkers associated with brain injury, including Tau proteins linked to Alzheimer’s pathology. Studies in brain organoids revealed that SARS-CoV-2 alters the phosphorylation and distribution of Tau in infected neurons, but the mechanisms are currently unknown. We hypothesize that these pathological changes are due to the recruitment of Tau into stress granules (SGs) operated by the nucleocapsid protein (NCAP) of SARS-CoV-2. To test this hypothesis, we investigated whether NCAP interacts with Tau and localizes to SGs in hippocampal neurons in vitro and in vivo. Mechanistically, we tested whether SUMOylation, a posttranslational modification of NCAP and Tau, modulates their distribution in SGs and their pathological interaction. We found that NCAP and Tau colocalize and physically interact. We also found that NCAP induces hyperphosphorylation of Tau and causes cognitive impairment in mice infected with NCAP in their hippocampus. Finally, we found that SUMOylation modulates NCAP SG formation in vitro and cognitive performance in infected mice. Our data demonstrate that NCAP induces Tau pathological changes both in vitro and in vivo. Moreover, we demonstrate that SUMO2 ameliorates NCAP-induced Tau pathology, highlighting the importance of the SUMOylation pathway as a target of intervention against neurotoxic insults, such as Tau oligomers and viral infection.

## 1. Introduction

While initially recognized for its respiratory effects, emerging research suggests that SARS-CoV-2 can affect various organs, including the central nervous system (CNS) [1]. Patients with COVID-19 commonly experience acute neurological symptoms like changes in smell and taste, along with persistent cognitive issues such as fatigue, headaches, attention problems, difficulty breathing, and cognitive decline (referred to as “brain fog”). This neurological aspect, termed “neuro-COVID” poses a significant global health concern, affecting even those with mild COVID-19 symptoms [2]. The underlying mechanisms behind this post-viral syndrome are multifaceted, potentially involving factors like reduced oxygen levels [3], disruption of the blood–brain barrier, and, notably, direct invasion [4] and damage to neuronal tissues by SARS-CoV-2 [5].

Recent evidence confirms the virus’s ability to target the nervous system, causing structural changes in the brain and metabolic alterations in survivors [6]. Ongoing studies are beginning to explore the link between COVID-19 and neurodegenerative conditions, suggesting that the virus may worsen existing dementia or predispose individuals to Alzheimer’s disease (AD) [7]. Notably, COVID-19 patients with neurological symptoms show elevated levels of biomarkers associated with brain injury, including Tau proteins linked to Alzheimer’s pathology [8,9].

Despite these findings, the extent to which SARS-CoV-2 directly infects neurons or indirectly affects them through non-neuronal cell infection remains uncertain. A study in brain organoids revealed that SARS-CoV-2 alters the phosphorylation and distribution of Tau in infected neurons [10], but the mechanisms are currently unknown.

We hypothesize that these pathological changes are due to the recruitment of Tau into stress granules (SGs) operated by the nucleocapsid protein (NCAP) of SARS-CoV-2.

The NCAP of coronaviruses plays a crucial role in the viral life cycle [11]. NCAP primarily binds to viral RNA, forms the viral nucleocapsid, and facilitates viral genome replication and transcription [12]. Studies have suggested that the nucleocapsid protein of coronaviruses, including SARS-CoV-2, may interact with SGs [13,14]. SGs are cellular structures that form in response to various stress conditions, such as viral infection [15], oxidative stress, or heat shock [16]. The interaction between the NCAP and SGs may influence cellular processes, including the host cell’s ability to mount an effective antiviral response.

Interestingly, SUMOylation has emerged as an essential pathway that is manipulated by viruses to modulate antiviral responses [17], viral replication and pathogenesis [18], and the modulation of SG dynamics [19,20]. The process of SUMOylation is a multi-step cascade, where a Small Ubiquitin-like MOdifier (SUMO) is covalently attached to a conserved ΨKxD/E motif within a target protein, altering the function of the modified protein [21,22]. There are three known SUMO paralogs in the brain: SUMO1, -2, and -3, with SUMO1 contributing to tissue damage and SUMO2/-3 ameliorating it [23,24,25,26]. Interestingly, the nucleocapsid of SARS-CoV is a target of SUMOylation [27]. SUMO is conjugated to NCAP at residue 62, and this modification triggers NCAP multimerization [28], a process relevant to binding to RNA for viral packaging [29].

Here, we tested whether SARS-CoV-2 infection and, in particular, the presence of high levels of NCAP may initiate or accelerate neurodegenerative processes, including those involved in AD, as observed for other pathogens [30,31] and suggested by some neurological symptoms that accompany COVID-19 [32,33].

To test whether exposure to SARS-CoV-2 may predispose to developing dementia, we investigated whether NCAP interacts with Tau and induces pathological changes in cell lines, human iPSC-derived cortical neurons, and mouse hippocampal neurons in vivo. Mechanistically, we tested whether posttranslational modification of NCAP and Tau by SUMO2 modulates their distribution in SGs and their pathological interaction.

Our data demonstrate that NCAP induces pathological changes in human neurons and animals expressing wild-type Tau and aggravates the pathology in neurons expressing mutant forms of Tau, both in vitro and in vivo. Moreover, we demonstrate that SUMO2 conjugation ameliorates NCAP-induced Tau pathology and directly reduces NCAP granules.

Our results suggest that SARS-CoV-2 may trigger molecular processes that disrupt Tau’s normal function, potentially contributing to neurological complications observed in COVID-19 patients. Our work also highlights the importance of the SUMOylation pathway as a target of intervention against neurotoxic insults, such as Tau oligomers and viral infection.

## 2. Results

### 2.1. NCAP Induces Formation of RNA Granules in Mouse Neuronal Cells, and SUMOylation Reduces It

SUMOylation has emerged as an essential pathway that is manipulated by viruses to modulate antiviral responses, viral replication, and pathogenesis [34]. The NCAP of SARS-CoV1 is a target of SUMOylation at the K62 residue, where the conjugation of SUMO1 promotes its oligomerization [27], and, more recently, the NCAP of SARS-CoV-2 was also reported to be a target of SUMO3 at the K65 residue [35]. Like the NCAP from SARS-CoV1, SUMOylation is also critical for SARS-CoV-2 NCAP self-association and phase-separated condensate formation [36], affecting NCAP activity.

Here, we tested whether the pharmacological modulation of SUMOylation resulted in changes in NCAP phase-separated condensate formation. First, we expressed NCAP-GFP in N2a mouse neuronal cells and found that NCAP-GFP had a mostly diffused appearance, with only one or a few condensates per cell (Figure 1A). To reduce or promote SUMO conjugation, we treated N2a cells expressing NCAP-GFP with either Ginkgolic acid [37] (10 µM) or Ebselen [38] (2 µM), respectively, for 16 h before imaging them. We found that the treatment with Ginkgolic acid significantly increased the number of condensates per cell (one-way ANOVA, *p* < 0.0001). At the same time, Ebselen significantly reduced the number, suggesting that a global increase in SUMOylation reduces the formation of NCAP condensates. Therefore, from these experiments, we hypothesized that an increase in global SUMOylation by reducing NCAP condensates could be beneficial in cells infected by SARS2-CoV2.

### 2.2. NCAP Induces Formation of RNA Granules in Human Glutamatergic Neurons of AD Patients, and SUMO2 Reduces It

Next, we sought to confirm the effects of NCAP on SG formation using iPSC-derived neurons from AD patients carrying the A246E mutation in presenilin1. Using established protocols, we differentiated iPSCs into glutamatergic neurons for 30 days [39]. At 30 DIV, the cells were transfected with NCAP-GFP and fixed 5 days later to visualize NCAP. We confirmed that NCAP-GFP induced the formation of condensates in the human neurons (Figure 1B), as previously observed in the N2a cells (Figure 1A). These results suggest that NCAP has the ability to induce the formation of SGs in human neurons.

Since our pharmacological treatments in the N2a cells demonstrated that an increase in SUMO conjugation reduced the number of NCAP condensates, we tested whether increased levels of SUMO2 could reproduce the same effects in iPSC-derived human neurons. At 30 DIV, neurons were transfected with NCAP-GFP alone or with SUMO2 and fixed 5 days later to visualize NCAP-GFP condensates. We found that the NCAP-SUMO2-transfected neurons had a significant reduction (*p* < 0.05 vs. NCAP alone) in the size of condensates and a concomitant increase in the amount of diffused signal (Figure 1B), suggesting that the modulation of SUMOylation can affect the distribution of NCAP in human neurons.

### 2.3. Wild-Type and Mutant Tau Physically Interact with NCAP

NCAP localizes to SGs [13], biomolecular condensates where Tau was also found to reside [40,41]. The presence of Tau in these organelles is now considered a critical factor in triggering its aggregation [42]. To address whether exposure to SARS-CoV-2 may predispose to developing dementia, we investigated whether NCAP colocalizes and interacts with Tau.

First, we transfected HEK293 cells with NCAP-GFP and found that NCAP induced the formation of molecular condensates that resembled SGs (Figure 2A). To determine the nature of these condensates, 48 h after transfection, we fixed HEK293 cells and stained them with an antibody to detect the stress granule marker G3BP1 [43]. Using confocal microscopy, we found that the G3BP1 colocalized with NCAP (Figure 2A). Next, we co-transfected HEK293 cells with NCAP-GFP plus Tau-Dsred. The cells were fixed and stained with anti-G3BP1 antibodies two days after transfection to determine whether NCAP induced the formation of SGs that recruited Tau. We found that Tau colocalized with NCAP-GFP in condensates (Figure 2B), suggesting that NCAP can recruit Tau into SGs.

To further demonstrate that NCAP and Tau interact, we performed co-immunoprecipitation (IP) experiments on lysates of HEK293 cells expressing NCAP-GFP and Tau-HA, both wild-type and carrying the human mutation P301L, which is associated with frontotemporal dementia [44]. Cells were transfected with NCAP alone, Tau alone, or with both Tau and NCAP. Forty-eight hours after transfection, cells were lyzed, and cleared lysates were immunoprecipitated with GFP magnetic beads to pull down NCAP-GFP. Immunoprecipitated proteins were visualized using anti-GFP (for NCAP, Figure 2C) and anti-HA (for Tau, Figure 2D). Wild-type and mutant Tau were co-immunoprecipitated with NCAP, suggesting that mutations on Tau do not affect its ability to interact with NCAP.

### 2.4. Establishing a Model of NCAP Overexpression in the Absence of SARS-CoV-2 Infectivity

A study in brain organoids revealed that SARS-CoV-2 alters the phosphorylation and distribution of Tau in infected neurons [45]. After establishing that NCAP and Tau colocalized into molecular condensates (SG) and that SUMO2 significantly reduced the number of NCAP-induced condensates, we next sought to test whether NCAP could induce pathological changes in Tau in vivo and whether SUMO2 might reduce them. However, infecting mice with competent SARS-CoV-2 particles requires special facilities and safety protocols that are not easily accessible for all research groups. Therefore, to assess the impact of NCAP on Tau in vivo, we opted for expressing only the gene of NCAP and not the complete SARS-CoV-2 genome. To achieve this, we infected JNPL3 mice [46], an animal model of frontotemporal dementia (FTD) carrying the human P301L mutation in the Tau gene, and aged-matched C57 wild-type animals (WT) with adeno-associated viruses (AAVs) to express NCAP alone or in combination with SUMO2.

Five-month-old JNPL3 and WT mice with six animals per group (three males and three females) were infected with GFP, NCAP-GFP alone, or NCAP-GFP+SUMO2-RFP viruses in the dorsal hippocampus (Figure 3A). Three months after the injection of the AAVs, the mice were sacrificed, and brains were collected to perform immunohistological analyses and Western blotting (Figure 3B). We found a robust expression of NCAP protein in the mice injected with NCAP-GFP alone and with NCAP-GFP+SUMO2 checked by immunofluorescence with an anti-NCAP antibody, and we detected its hippocampal distribution, together with that of RFP, used as reporters for SUMO2 AAVs (Figure 3B). NCAP and SUMO2 expression was also confirmed by Western blotting (Figure 3D). In particular, we observed an increase in SUMO2 conjugation in the mice injected with NCAP-GFP+SUMO2 viruses compared with the mice injected with either GFP or NCAP alone (Figure 3E).

### 2.5. NCAP Induces an Increase in Tau Phosphorylation in Vivo, and SUMO2 Reduces It

Next, we assessed if NCAP could induce an increase in the phosphorylation of Tau in the hippocampus of WT and JNPL3 mice. We performed immunofluorescence analyses using antibodies specific to detecting NCAP and phosphorylated Tau at the pathological residues Ser 202 and 205 (AT8), which are associated with AD [15]. We found some AT8-positive cells in the WT mice expressing NCAP (Figure 4A), and the co-expression of SUMO2 reduced the AT8-positive signal. JNPL3 mice already have AT8-reactive neurons in their hippocampus, and NCAP increased the AT8 signal. As for the WT mice, SUMO2 reduced the amount of the AT8 signal in the hippocampus of the JNPL3 mice injected with NCAP. We confirmed these results by performing quantitative Western blotting analyses (Figure 4B,C).

Phosphorylation of Tau is accompanied by cognitive and motor disturbances in animal models of Tauopathies and in AD and FTD patients. Therefore, we next tested whether NCAP could induce cognitive and motor defects in WT and JNPL3 mice.

### 2.6. SUMO2 Improves Motor and Cognitive Deficits Induced by NCAP in WT and JNPL3 Animals

Our hypothesis was that infection with SARS-CoV-2 may have detrimental effects in the brain and induce neurodegenerative diseases in otherwise healthy subjects or aggravate neurodegeneration in patients already affected by AD or FTD. To test our hypothesis, we assessed motor and cognitive performances in WT and JNPL3 mice injected with GFP, NCAP, or co-injected with NCAP and SUMO2.

Rotarod was performed six weeks (1.5 months) and 3 months after AAV injection (Figure 5A) to assess motor performance. Overall, the JNPL3 mice performed significantly worse than the WT mice (two-way ANOVA; *p* = 0.0001 and *p* = 0.0003 for 1.5 and 3 months respectively; Figure 5A). However, post hoc analyses showed that the JNPL3 mice infected with GFP were not significantly different from the WT mice at both time points. Instead, NCAP expression in the JNPL3 mice’s hippocampus significantly worsened their performance (one-way ANOVA; *p* = 0.003 and *p* = 0.005 for 1.5 and 3 months respectively; Figure 5B). The co-expression of SUMO2 did not significantly improve the performance 1.5 months after infection. However, three months after infection, the JNPL3 mice expressing SUMO2 performed like the mice expressing GFP (Figure 5B).

The animals were also trained in a Y-maze two months after infection to assess their working memory. In this task, the mice were allowed to explore only one arm of the maze during the training phase, while they were permitted to explore all the arms during the testing phase, which was performed twenty-four hours after training. Cognitively competent mice tend to spend more time and explore first the novel arm. As expected, the JNPL3 mice performed worse than the WT mice (Figure 5C), as demonstrated by the difference in the discrimination index (DI, two-way ANOVA, *p* = 0.025). Infection with NCAP did not significantly change the DI of the JNPL3 mice, which was already extremely poor. We also examined the frequency of exploring the new arm first (Figure 5D). We found that expression of NCAP in the WT mice significantly reduced the number of mice that chose to explore the new arm first (two-way ANOVA; genotype effect: *p* = 0.0285, and AAV effect: *p* = 0.0187; Figure 5D). At the same time, the co-expression of SUMO2 rescued the deficits.

In addition, we also measured the latency to enter the new and old arms, another parameter used to assess the preference of mice toward the new arm. Mice with intact memory will go first to the new arm and, hence, will have lower latency to explore that arm, while they will have longer latency to go to the old arm. This analysis revealed more differences between the different groups. We found that NCAP significantly reduced the performance of the WT animals compared with the WT animals injected with GFP, and we found that that SUMO2 overexpression rescued the cognitive impairment caused by NCAP in the WT mice (two-way ANOVA; AAV effect: *p* = 0.018, arm effect: *p* = 0.0006; Figure 5E). Moreover, the JNPL3 mice infected with GFP were already severely impaired, and NCAP did not significantly worsen their performance, but SUMO2 significantly improved it (two-way ANOVA; AAV effect: *p* = 0.0341, arm effect: not significant; Figure 5E), with post hoc multiple comparison analyses revealing a significant difference between the new and old arm only when the JNPL3 mice were injected with NCAP and SUMO2 (*p* = 0.035).

Together, these data suggest that NCAP can induce cognitive impairment in WT animals and worsen the cognitive impairment already present in JNPL3 mice and that increasing SUMO2 conjugation can improve or restore cognition.

## 3. Discussion

The main findings of our study are that [1] NCAP expressed in the hippocampal neurons of WT as well as JNPL3 mice induced an increase in Tau phosphorylation, resulting in cognitive impairment, and that [2] an increase in the neuroprotective SUMO2 conjugation improved the behavioral outcomes in the models studied. This protection may be explained by the fact that an increase in SUMO2 conjugation significantly reduced the number and size of NCAP condensates.

Reducing the number of NCAP condensates could be beneficial in the context of viral infection to counteract NCAP’s ability to inhibit host cell signaling for evading innate antiviral immunity, which requires the formation of NCAP condensates.

One of the host responses to arresting viral replication is to form SGs [15], transient cytoplasmic aggregates of mRNA and proteins induced to stop the translation of viral RNAs [16]. To counteract the host response, viruses can sequestrate SG components to promote viral replication [15]. In this context, the ability of NCAP to sequestrate SG components into condensates could be beneficial for the virus.

The induction of SGs is particularly relevant in neuronal cells, since SGs are also emerging as critical players in neurodegenerative diseases such as Alzheimer’s, Parkinson’s, and amyotrophic lateral sclerosis (ALS) [47,48,49]. Several essential proteins associated with these diseases, such as TDP-43, FUS, and Tau, have been found to localize to SGs under stress conditions. Dysregulation of SG dynamics, including aberrant assembly, persistence, or inefficient clearance, may contribute to neuronal dysfunction and degeneration in these diseases [41,43]. One proposed mechanism is that chronic SG formation could lead to the sequestration of essential RNA-binding proteins and mRNAs, disrupting normal cellular function and potentially contributing to the accumulation of protein aggregates characteristic of neurodegenerative diseases. Also, SGs may impair the clearance of misfolded proteins or exacerbate cellular stress responses, compromising neuronal viability. Therefore, SGs could be a potential juncture between SARS-CoV-2 infectivity and neurodegeneration, and their modulation could offer therapeutic implications.

Interestingly, SUMOylation has emerged as an essential pathway that viruses manipulate to modulate antiviral responses, viral replication and pathogenesis [18], and the modulation of SG dynamics [17].

Several stress-granule-associated proteins are SUMOylated, including RNA-binding proteins such as TIA-1, G3BP1, and hnRNPA1. The SUMOylation of these proteins can impact their localization to SGs, their interactions with other granule components, and their ability to regulate mRNA metabolism and translation within SGs. Furthermore, SUMOylation enzymes, including SUMO E3 ligases and SUMO-specific proteases, have been shown to regulate SG formation and disassembly [19]. For example, SUMO-specific proteases such as SENP3 have been found to regulate SG disassembly by deSUMOylating SG components [19].

Therefore, dysregulation of SUMOylation pathways may impact SG formation and clearance [50], potentially influencing cellular homeostasis and contributing to disease states such as neurodegeneration. Thus, acting on the SUMO pathway could be beneficial in reducing the NCAP-mediated recruitment of SG components.

Our data support this hypothesis. We found that increasing SUMO2 protein and pharmacological treatment with Ebselen to increase SUMO2 conjugation resulted in the disassembly of NCAP condensates. Moreover, overexpression of SUMO2 significantly improved cognitive function in the mice infected with NCAP.

In contrast with our findings, direct SUMOylation of NCAP has been recently proposed to promote NCAP oligomerization and condensate formation, suggesting that SUMOylation of NCAP could be detrimental to infected cells [36].

In fact, the NCAP of SARS-CoV, the first of the new coronaviruses, has already been reported to be a target of SUMOylation [28]. In particular, isoform SUMO1 was found to trigger NCAP oligomerization, a process relevant to binding to RNA for viral packaging.

Interestingly, previous work has focused on the effects of SUMO1 on NCAP oligomerization. While we did not directly test the effects of increasing SUMO1, it has been reported by others that different SUMOs can have different effects on their target proteins [51]. It is therefore conceivable that while SUMO1 increases NCAP oligomerization, SUMO2 instead reduces it. It is also possible that the reduction in NCAP condensates we observed with increased levels of SUMO2 originates from the effects of SUMO2 on other targets and not on NCAP.

Our data in fact suggest that a general increase in SUMO2 might overcome the NCAP-specific effects. Overexpression of SUMO2 results in the conjugation of SUMO2 to different targets, possibly hundreds, and the global impact of targeting so many other proteins could compensate for the changes occurring only on NCAP.

An important aspect of our work is that NCAP expressed in the hippocampus induced cognitive impairments in the WT mice. These data support our original hypothesis that infection with SARS-CoV-2 might in fact predispose individuals to develop dementia after contracting the virus. Recent epidemiological data further support our conclusions [52]. Another important aspect of our findings is that NCAP worsened the cognitive impairment already present in the JNPL3 mice. This may suggest that patients that are already affected by neurodegenerative diseases could experience a worsening of their conditions after being exposed to SARS-CoV-2 and developing COVID-19.

Mechanistically we found that the expression of NCAP in the hippocampus was associated with an increase in Tau phosphorylation. Our findings are in agreement with data published by others [10,53] in brain organoids infected with SARS-CoV-2. Interestingly, we also found that expression of SUMO2 was accompanied by a reduction in the phosphorylation of Tau. How does SUMO2 affect Tau phosphorylation? It is possible that one or more SUMO2 targets influence Tau phosphorylation. For example, GSK3β, the kinase that phosphorylates Tau on residues 202 and 205, is a target of SUMOylation, and the SUMOylation state of GSK3β is associated with its kinase activity [54]. Thus, it is possible that by altering GSK3β activity, SUMO2 may indirectly alter Tau phosphorylation. SUMO2 could also act directly on Tau. Tau in fact is a target of SUMOylation [55], and SUMOylation of Tau has been reported to modulate its phosphorylation [56]. Therefore, SUMO2 might counteract the NCAP-induced increase in Tau phosphorylation acting on multiple targets.

A limitation of our work is that we utilized recombinant AAVs to induce expression of NCAP directly into the hippocampus instead of infecting the mice with the full SARS-CoV-2 virus. Our approach could potentially induce the presence of much higher levels of NCAP in the brain compared to the amount that could originate from peripheral exposure to the virus. However, our reductionist approach also offers several advantages. For example, we can assess the specific effects caused by the expression of NCAP alone compared with the expression of the whole viral genome; we can also establish the direct effects of NCAP onto Tau expressed in the hippocampus without the confounding effects of inflammation caused by SARS-CoV-2, which results in altering the blood–brain barrier, both of which have been previously associated with changes in Tau pathology.

In conclusion, our data demonstrate that NCAP induced pathological changes in the hippocampal neurons of animals expressing wild-type Tau and aggravated the pathology in neurons expressing mutant forms of Tau, both in vitro and in vivo. Moreover, we demonstrated that SUMO2 conjugation ameliorated NCAP-induced Tau pathology and directly reduced NCAP granules in mouse neurons and human iPSC-derived neurons carrying an AD-related mutation in PSEN1.

Overall, our work highlights the importance of the SUMOylation pathway as a target of intervention against neurotoxic insults, such as pathological forms of Tau and, more in general, viral infection.

## 4. Materials and Methods

### 4.1. N2a and HEK293 Cell Culture

HEK293 and N2A cells obtained from ATCC (American Type Culture Collection, Manassas, VI, USA) were cultured in Dulbecco’s modified medium (DMEM) with 10% fetal bovine serum (FBS) and 100 U/mL penicillin–streptomycin and incubated at 37 °C with 5% CO_2_.

### 4.2. HIPSC-Derived Neurons

PSEN1 A246E-mutated neural progenitors (NPCs) were purchased from Axol Bioscience (Product code AX0114). NPCs were grown and expanded according to the manufacturer’s instructions and maintained in Neural Progenitor Medium (Stem Cell Technologies, Vancouver, BC, Canada) on Matrigel (Corning, Corning, NY, USA)-coated wells up to the fifth passage before being detached and seeded on poly-L-lysine-coated glass coverslips at 50,000–60,000 cells/cm^2^. Cells were then differentiated by replacing the maintenance medium with BrainPhys Media (Stem Cell Technologies, Vancouver, BC, Canada) after 24 h from the seeding up to 35 days in vitro.

### 4.3. Plasmids for Transient Expression

NCAP-GFP (VG40588-ACG) was purchased from Sino biological, and Tau GFP and TauP301L GFP were obtained from Addgene (Plasmid #46904, Plasmid #46908). To obtain HA-tagged Tau, Tau GFP and TauP301L GFP were subcloned into pcDNA3.1+ SUMO2-CPEB3 [57], replacing CPEB3 using the following primers: forward primer 5′-CTAGGGATCCAATGGCTGAGCCCCGCCAG-3′; reverse primer 5′-CTAGCTCGAGTTAAGCGTAATCTGGAACATCGTATGGGTAAGCAGCCAAACCCTGCTTGGC-3′). Tau with a red fluorescent tag was purchased from Addgene mCherry-MAPTau-C-10 (Plasmid #55077). Cell lines and hiPSC were transfected using Lipofectamin™ 3000 (#L3000-015, Invitrogen, Waltham, MA, USA) diluted in serum-free media. A total of 500 ng of DNA per 24-well culture dish or 2 µg per 6-well culture dish were used.

### 4.4. Chemicals and Antibodies

Ebselen and Ginkgolic acid were purchased from Tocris (Bristol, UK).

The primary and secondary antibodies used for the specific applications are listed in Table 1 and Table 2, respectively.

### 4.5. Immunoprecipitation

Cells were homogenized in lysis buffer (0.5% Triton X-100; 0.5% NCAP-40 and 50 mM Tris-HCl (pH 7.5)) with N-Ethylmaleimide (NEM), 10 mM phenylmethylsulfonyl fluoride, and complete protease inhibitor cocktail (Roche Applied Science, Basel, Switzerland). The homogenates (300 µg) were incubated for 1 h at 4 °C with Chromotek GFP magnetic beads. The beads were washed four times with washing buffer for 5 min at 4 °C. Bound proteins were eluted with Laemmli sample buffer and analyzed by Western blotting.

### 4.6. Animals and Adeno-Associated Viral Transduction

Procedures involving animals were conducted in conformity with the institutional guidelines at the Istituto di Ricerche Farmacologiche Mario Negri IRCCS in compliance with national (D.lgs 26/2014; authorization no. 19/2008-A, issued 6 March 2008, by the Ministry of Health) and international laws and policies (EEC Council Directive 2010/63/UE; the NIH Guide for the Care and Use of Laboratory Animals, 2011 edition). The animal facilities met international standards and were regularly checked by a certified veterinarian who was responsible for health monitoring, animal welfare supervision, experimental protocols, and reviewing of procedures. Homozygous JNPL3 mice were purchased from Taconic (Rensselear, NY, USA), strain ID #2508. JNPL3 and WT mice (Envigo, Bresso, Italy), 5-months old, both males and females, were anesthetized with 3% isoflurane in N2O/O2 (70/30%) and maintained by from 1.5 to 2% isoflurane in the same gas mixture during the stereotactic surgery. Intrahippocampal injections of 1.3 × 10^9^ infective units (IU) of either GFP, NCAP alone, or in combination with the same units of SUMO2 were performed at the following coordinates: posterior −1.95 mm, lateral ±1.4 mm, and ventral −2.0 and −1.6 mm. In each injection site, 1 µL of viral suspension was infused using a rate of 0.5 µL/min, and the needle was left in place for 3 min before withdrawal. The AAV constructs used were the following: AAV1-CMV-RFP-CMV-hSUMO2, custom-made and purchased from Vector Biolabs (Malvern, PA, USA); NCAP-GFP YP_009724397.2 GeneCopeia (#NV230); and control AAV1-GFP.

### 4.7. Immunofluorescence Analyses on Hippocampal Sections

Mice were perfused with 20 mL of ice-cold phosphate-buffered saline (PBS) at 0.1 M and pH 7.4. The brains were removed from the skull and post-fixed in 10% formalin. Then, the tissues were dehydrated in graded ethanol solutions, cleared in xylene, and embedded in paraffin. Serial sagittal formalin-fixed paraffin-embedded (FFPE) sections (8 µm thick) were cut on a microtome (Leica Biosystems, Wetzlar, Germany). The slices were first deparaffined in a heater at 58 °C for 20 min, then washed in serial dilutions in a decreasing ethanol scale, and lastly washed in deionized water. For heat-induced antigen retrieval, sections were incubated in sodium citrate buffer at pH 6.0 (Antigen Decloaker 10x, Bio Optica, Milano, Italy, CB910M) in a microwave for 5 min at 750 Watts. After blocking with 3% bovine serum albumin (BSA) in TBST-T (0.1% Triton X-100 in TBS (Tris-Buffered Saline, 0.01 M), the sections were incubated at 4 °C overnight in 1% BSA with 0.1% Triton X-100 in TBS containing the following primary antibodies (see Table 1): rabbit anti-SARS-CoV-2 (COVID-19) nucleocapsid antibody and mouse AT8 anti-phospho-Tau (Ser202, Thr205). After washing with TBS, the sections were incubated with secondary antibody solution containing 1% BSA in TBS-T for 1 h at RT (see Table 2). The nuclei were stained with DAPI (2 µg/mL). The sections were rinsed and covered with Prolong gold anti-fade reagent (Invitrogen, Waltham, MA, USA).

### 4.8. Confocal Acquisition

Fluorescent images were acquired using a confocal A1 system (Nikon, Tokyo, Japan) equipped with a confocal scan unit with 405 nm, 488 nm, 561 nm, and 640 nm laser lines with a scanning sequential mode to avoid bleed-through effects. For the cells, three-dimensional and higher-magnification images were acquired using a 60× water immersion objective over an 8- to 10-µm *z* axis with a 0.273 µm step size and processed by using the Imaris software 7.4.2 (Bitplane). For the AAV distributions, the hippocampi were evaluated by confocal analysis, and images were acquired by stitching adjacent 20× frames. For higher magnification, the images were acquired using a 60× water immersion objective over an 8- to 10-µm *z* axis with a 0.225 µm step size (resolution: 0.21 µm/px).

### 4.9. NCAP Granule Quantification

To quantify NCAP granules, 3–5 images totaling >100 cells per sample were acquired using a 40× objective. The granules were analyzed automatically with the Image J plugin “Particle analysis”. A random selection of images was hand-verified. The NCAP granules average size were analyzed by the Image J software version 1.53C using the ‘analyze particles’ instrument with the threshold “size = 0.00–100.00”.

### 4.10. Protein Extraction from Mouse Hippocampi

For the biochemical analysis, the mice were perfused with 20 mL of ice-cold PBS, the brains were dissected to separate the hippocampus, cortex, and striatum, and they were snap-frozen. The hippocampi were homogenized in extraction buffer (50 mM Tris-HCl [pH 7.5], 50 mM KCl, and 10 mM MgCl2) supplemented with complete protease and phosphatase inhibitor cocktails (Roche Applied Science) and N-Ethylmaleimide (NEM) at 20 mM. Cell debris was removed by centrifugation at 7000× *g* rpm for 10 min.

### 4.11. Western Blot

Immunoprecipitated samples and hippocampal lysates (20 µg) were run in 4–20% gradient Tris-glycine TGX™ Precast Stain Free Protein Gels (Bio-Rad, Hercules, CA, USA) and transferred onto nitrocellulose or low-fluorescence PVDF membranes using a semi-dry Trans-Blot^®^ Turbo™ Transfer System (Bio-Rad). The membranes were blocked in 3% BSA for 1 h at RT and then probed with mouse anti-HA antibody, chicken anti-total tau, and rabbit anti-SUMO2 (see Table 1) overnight at 4 °C followed by HRP or fluorescent secondary antibodies (see Table 2). Western blot images were acquired with a ChemiDoc MP Imaging System (Bio-Rad, Hercules, CA, USA) and quantified using the Image lab software 6.0.

### 4.12. Behavioral Studies

Rotarod: The mice were trained to perform the test by placing them on a rotating bar (Ugo Basile, Gemonio, Italy) with a controlled speed and acceleration (10 rpm for 60 s) (habituation and training phase); if the animal fell, the trial was repeated up to a maximum of 3 times. Once trained, the mice performed the test that was composed of a total of 3 trials with an inter-trial interval of 30 min. Each mouse was allowed to walk on the rotating bar for up to 300 s per trial with acceleration from 4 up to 40 rpm. The latency to fall from the bar was recorded, and the average over the 3 trials was calculated.

Y-maze: The Y-maze apparatus consisted of 3 arms spaced 120° apart to form a “Y” (width 5 cm × length 35 cm × height 20 cm), with spatial cues on each internal. The test took place in two phases: an acquisition phase (trial 1), in which an arm of the maze, selected randomly, was closed with a guillotine door and the mouse was positioned in an open arm (defined as the ‘departure’ or ‘start arm’) and left free to explore both the starting arm and the other open one (called the ‘old arm’) for 5 min; and a test phase (trial 2), in which the mouse was positioned in the same starting arm as in trial 1 but was left free to explore all 3 arms for 5 min. The Ethovision XT, 15 software (Noldus Information Technology, Wageningen, Holland) was used to record the sessions and track the path of the mice over the Y-maze.

### 4.13. Statistical Analysis

Data are expressed as mean or median ± standard error of the mean (SEM) and graphed as scatter plots. The GraphPad Prism 9 software was used for the statistical analysis. All data were tested for normal distribution using the Shapiro–Wilk normality test. For >2 groups, the results were analyzed by one-way ANOVA followed by Tukey’s multiple comparisons test or two-way ANOVA followed by Sidak’s post hoc multiple comparisons test. In the case of =2 groups, the data were compared using the unpaired *t*-test since they were normally distributed. Additional information on the tests used is provided in the figure legends.

## Figures and Tables

**Figure 1 ijms-25-07169-f001:**
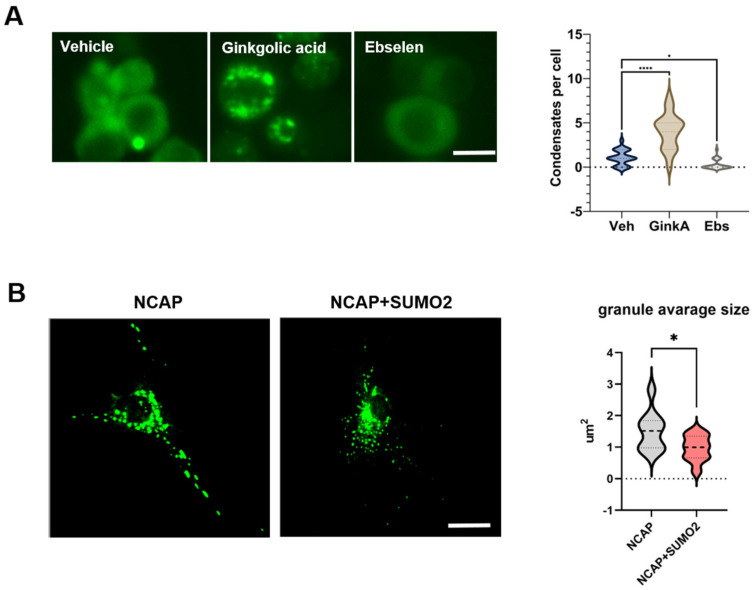
NCAP forms condensates that are modulated by SUMOylation in neuronal cells. (**A**) Quantification of NCAP condensates in N2a cells transfected with NCAP-GFP and treated with drugs that affect SUMOylation, i.e., Ebselen (2 µM) or Ginkgolic acid (10 µM), for 16 h. Modulators of SUMOylation altered the number of NCAP condensates in N2a compared with vehicle alone; one-way ANOVA followed by Tukey’s multiple comparisons test; * *p* < 0.05, **** *p* < 0.0001 (*n* = 69, from three independent replicates). Scale bar: 20 µm. (**B**) Neural progenitor cells (NPCs) were differentiated into glutamatergic neurons for over 30 days in vitro before transfection with NCAP-GFP and SUMO2. After 5 days from transfection, cells were fixed, and images of transfected NCAP were acquired by confocal microscope. Co-transfection with SUMO2 induced a significant reduction in the size of NCAP-positive granules, unpaired *t*-test: * *p* < 0.05 (*n* = 10 neurons per condition). Scale bar: 20 µm.

**Figure 2 ijms-25-07169-f002:**
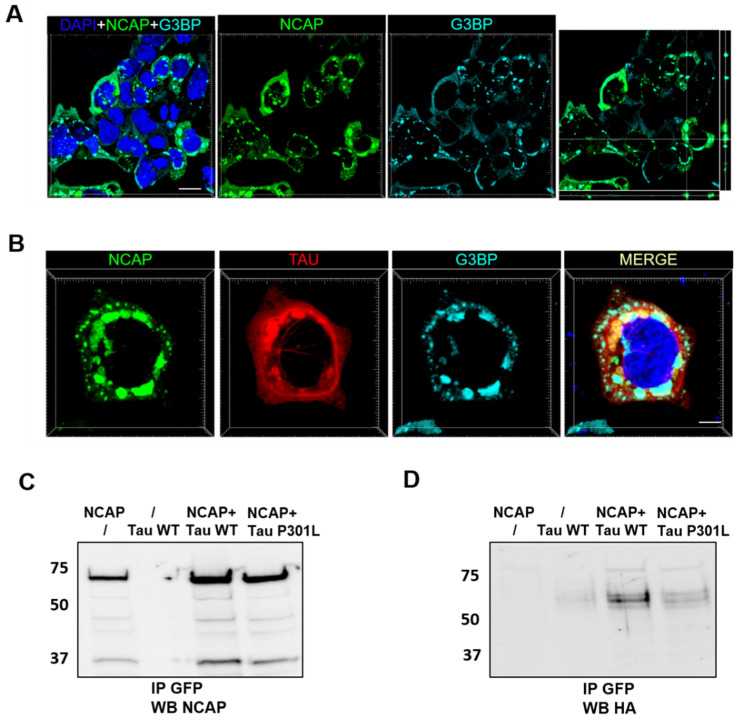
NCAP colocalizes with G3BP1 and interacts with Tau. (**A**) HEK293 cells were transfected with NCAP-GFP. A total of 48 h later, cells were fixed and probed with anti-G3BP1 antibody, and images were acquired by confocal microscopy. NCAP and G3BP1 colocalized in granules, as shown by x, y single-plane image with z-projections. Scale bar: 20 µm. (**B**) HEK293 cells were transfected with NCAP-GFP and Tau-RFP. A total of 48 h later, cells were fixed and probed with anti-G3BP1 as in (**A**). NCAP colocalized with both G3BP1 and Tau-RFP in granules; scale bar: 10 µm. (**C**) Representative WB of immunoprecipitated NCAP from HEK cells transfected with NCAP-GFP, Tau HA, or NCAP plus Tau-HA/TauP301L-HA. A total of 48 h after transfection, cells were lyzed, and cleared lysates were incubated with GFP magnetic beads to immunoprecipitate NCAP-GFP. IP proteins were visualized using anti-GFP antibody (for NCAP, panel (**C**)) and anti-HA (for Tau, panel (**D**)). Both wild-type and mutant Tau were co-immunoprecipitated with NCAP.

**Figure 3 ijms-25-07169-f003:**
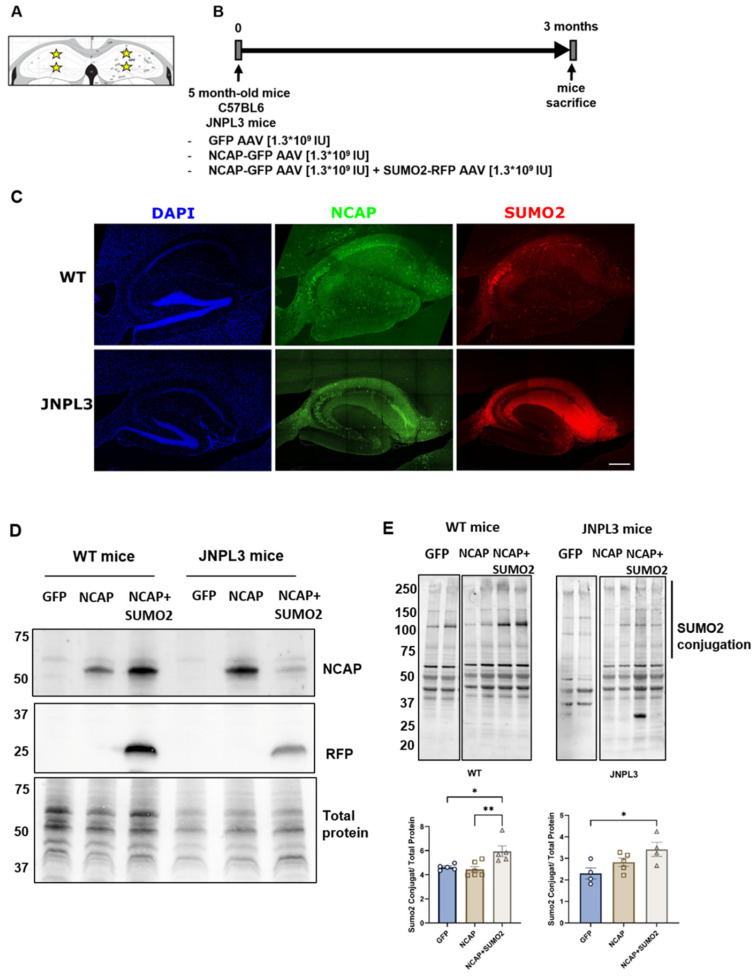
AAV model of NCAP injection in WT and JNPL3 mice. (**A**) Schematic representation of the experimental design showing the injection sites (yellow stars) and time of expression of AAVs. (**B**) Five-month-old wild-type mice (WT) and JNPL3 mice received hippocampal injections of AAV transducing NCAP-GFP (1.3 × 10^9^ IU) or NCAP-GFP (1.3 × 10^9^ IU) together with AAV transducing SUMO2 (1.3 × 10^9^ IU). AAV transducing GFP was used as control. NCAP was expressed together with GFP under the same promoter. SUMO2 was expressed together with RFP under a different promoter. (**C**) Representative images of transduced hippocampal slices: AAV NCAP distribution in green, AAV SUMO2 distribution in red, and nuclei (stained with DAPI) in blue; scale bar: 200 µm. (**D**) Western blot confirming the expression of NCAP and RFP. (**E**) Western blot and quantification confirming that SUMO2 conjugation was increased by AAV SUMO2 injection in transduced hippocampi of WT and JNPL3 mice. One-way ANOVA followed by Tukey’s test; * *p* < 0.05, ** *p* < 0.001. Lanes of GFP samples were from the same blot of NCAP and NCAP plus SUMO2 but were not contiguous, thus are separated with a vertical line from the other lanes.

**Figure 4 ijms-25-07169-f004:**
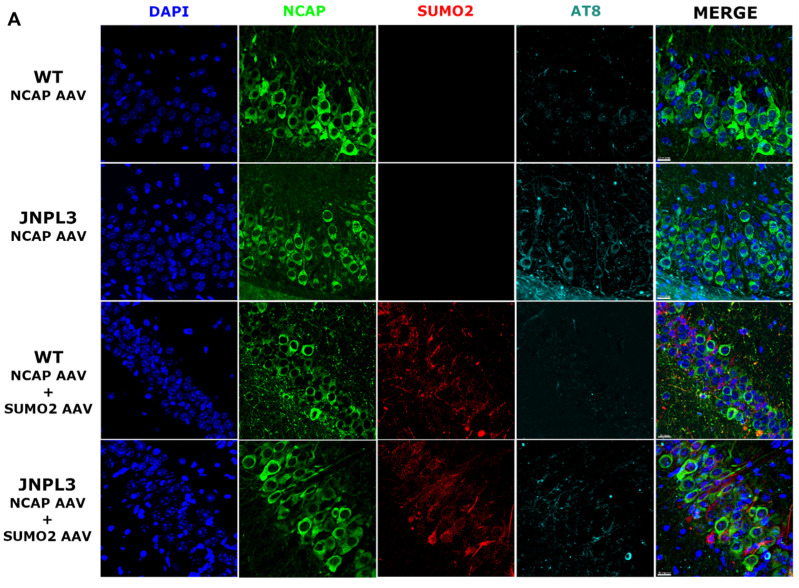
NCAP induces hyperphosphorylation of WT and P301L Tau in the hippocampus. (**A**) Immunofluorescence of representative hippocampal sections from WT and JNPL3 animals injected with NCAP (green) alone or in combination with SUMO2 (red) AAV in the dorsal hippocampus. Phosphorylated Tau at Serine 202 and 205 (AT8 antibody) is shown in cyan. (**B**) Representative images of Western blots of hippocampal protein extracts from WT and JNPL3 mice injected with GFP, NCAP alone, or NCAP and SUMO2. Top panels show total Tau, middle panels show phospho Tau (AT8), and bottom panels show a portion of total proteins. (**C**) Quantifications of Western blots; NCAP induced a significant increase in Tau phosphorylation compared with GFP in both WT and JNPL3 mice, and SUMO2 reduced phosphorylation of Tau; one-way ANOVA followed by Tukey’s multiple comparisons test; * *p* < 0.05, ** *p* < 0.01. Bars represent mean ± SEM of four mice represented by individual points.

**Figure 5 ijms-25-07169-f005:**
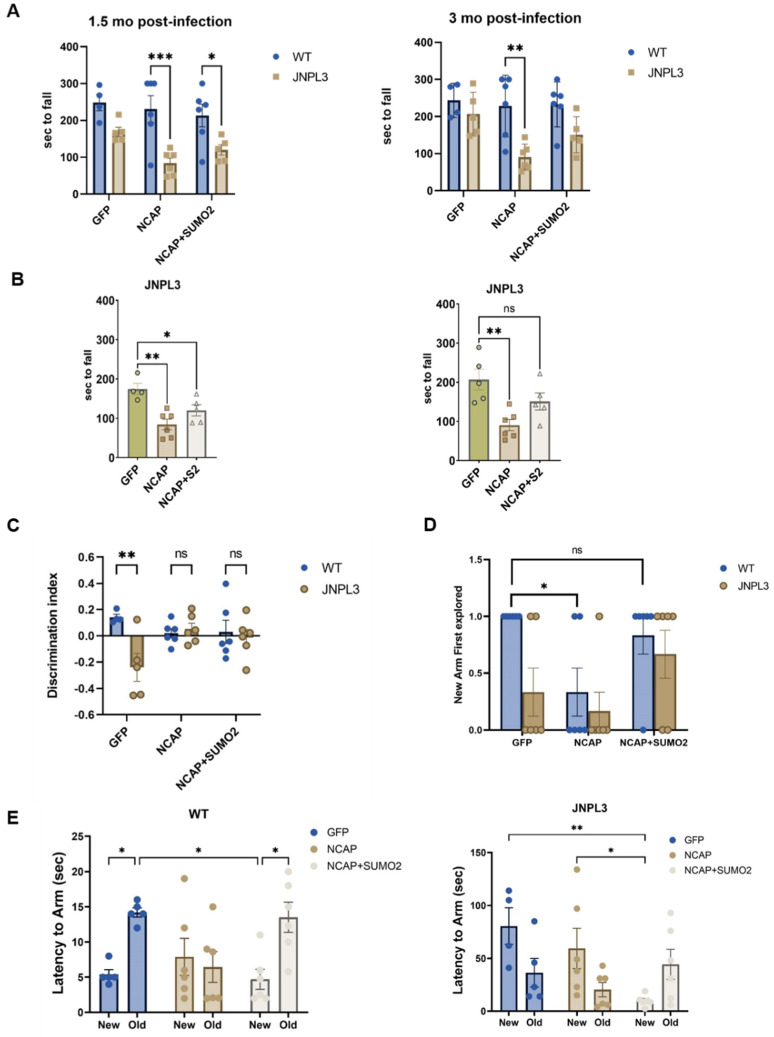
Behavioral assessments in WT and JNPL3 mice expressing NCAP and SUMO2 in the hippocampus. (**A**) Before the injections, 5-month-old WT and JNPL3 mice were trained on a rotating bar. 1.5 and 3 months after AAV injection in the dorsal hippocampus with GFP, NCAP alone, or NCAP+SUMO2, animals were tested for their motor coordination and balance by the rotarod test. Latency to fall (seconds) was recorded. Two-way ANOVA followed by Sidak’s test were conducted to compare genotypes; * *p* < 0.05, ** *p* < 0.01, and *** *p* < 0.001. (**B**) The graphs show the effect of the different AAVs within the JNPL3 genotype alone, and one-way ANOVA analysis showed that NCAP significantly worsened the performance of JNPL3 mice and SUMO2 restored it after 3 months but not after 1.5 months of expression (Tukey’s multiple comparisons test; * *p* < 0.05, ** *p* < 0.01). (**C**) The same animals were tested with a Y-maze 2 months after infections. Discrimination index of WT and JNPL3 mice injected with GFP showed that JNPL3 mice performed worse than WT animals and that NCAP induced a decline in WT mice. Two-way ANOVA followed by Sidak’s multiple comparisons post hoc test; ** *p* < 0.01. (**D**) The number of mice exploring the new arm first (value 1 on the y axis) and the old arm first (value 0 on the y axis) was also measured. All the WT mice injected with GFP had a score = 1, while injection with NCAP significantly worsened their performance, with only two out of six mice selecting the new arm first. SUMO2 improved this frequency significantly in both WT and JNPL3 mice. Two-way ANOVA followed by Sidak’s multiple comparisons test; * *p* < 0.05. (**E**) Latency to enter the new vs. old arm showed that NCAP severely impaired WT mice and worsened JNPL3 mice, while overexpression of SUMO2 improved memory loss. Two-way ANOVA followed by Sidak’s multiple comparisons test; * *p* < 0.05, ** *p* < 0.01. Bars represent mean ± SEM of 4–6 mice, represented by individual points.

**Table 1 ijms-25-07169-t001:** List of primary antibodies.

Primary Antibody	Host	Dilution for WB	Dilution for IF	Brand, Catalogue Number
G3BP1 (Clone 2F3)	mouse	-	1:250	Abcam, ab56574
GFP (9F9.F9)	mouse	1:1000	-	Abcam, ab1218
HA	rabbit	-	1:250	Abcam, ab9110
NCAP	rabbit	1:1000	1:500	Genetex, GTX35357
AT8	mouse	1:1000	1:40	Invitrogen, MN1020
RFP	rabbit	1:500	-	Abcam, ab2341
SUMO2	rabbit	1:1000	-	Abcam, 3742
Tau	chicken	1:5000	-	Invitrogen, PA5-95648

**Table 2 ijms-25-07169-t002:** List of secondary antibodies.

Secondary Antibody	Host	Dilution for WB	Dilution for IF	Brand, Catalogue Number
HRP	mouse	1:10,000		Jackson Immunoresearch, #115-035-174
HRP	rabbit	1:10,000	-	GE Healthcare, NA934
DyeLight 488	chicken	1:3000	-	Thermo Fisher SA5-10070
AlexaFluor 488	rabbit	-	-	Invitrogen, A-11008
AlexaFluor 594	rabbit	-	1:500	Invitrogen, SA5-10040
AlexaFluor 647	mouse	-	1:500	Invitrogen, A-21235
IRDye 680RD	rabbit	1:10,000	1:500	LI-COR Biscientific, 926-68073
IRDye 800CW	mouse	1:10,000	-	LI-COR Biscientific, 926-32212

WB = Western blotting, IF = immunofluorescence.

## Data Availability

All relevant data are contained within the article. The original contributions presented in the study are included in the article/Appendix A; further inquiries can be directed to the corresponding author.

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
