# Peer review of "SARS-CoV-2 Nucleocapsid Protein Induces Tau Pathological Changes That Can Be Counteracted by SUMO2"

_ijms, 2024, doi:10.3390/ijms25137169_

Round 1

Reviewer 1 Report

Comments and Suggestions for Authors

In this work, the authors address a topic that is highly topical and certainly of interest to the scientific community.

The authors highlighted that COVID-19 patients with neurological symptoms show elevated levels of biomarkers associated with brain injury, including Tau proteins linked to Alzheimer's pathology. It is highlighted that SARS-CoV2 alters the phosphorylation and distribution of Tau in infected neurons.

The authors investigated the possibility that the tau protein and the nucleocapsid protein of SARS-CoV2 could interact. In this work, the importance of post-translational modification of SUMOylation is discussed. This highlights the importance of the SUMOylation pathway as a target for action against neurotoxic insults, such as tau oligomers and viral infections.

Analyzing the paper in detail:

-Introduction:

This section is well written, the topics are explained clearly and in detail with excellent reference to the literature.

-Results:

 2.1.1. NCAP induces formation of RNA granules in mouse neuronal cells and SUMOylation reduces them. 

Has the SUMOylation of the protein at residues K62 and K65 also been verified in the present work by mass spectrometry analysis? If not, I think it might be interesting to see if the modifications are confirmed and if other amino acid residues are also modified.

2.1.5. NCAP induces an increase of Tau phosphorylation in vivo and SUMO2 reduces it. 

“We performed immunofluorescence analyses using antibodies specific to detect NCAP and phosphorylated Tau at the pathological residues Ser 202 and 205” 

Were the phosphorylation changes in the Tau protein recognised simply by using antibodies? Again, I would suggest analysing and verifying by protein digestion and mass spectrometry the phosphorylation sites.

-Discussion:

The conclusions presented by the authors are well rationalised and supported by the literature.  The possibility of the SUMOylation pathway as a target for intervention against neurotoxic insults, such as pathological forms of Tau and, more generally, viral infections, is introduced. This cue is innovative and may be of great interest to the scientific community. 

The article is worthy of publication in Int. J. Mol. Sci., in the present form, after clarifying some curiosities in the results section.

Reviewer 2 Report

Comments and Suggestions for Authors

Abstract: gives a good summary about the problem needs to be solved (biomarkers in COVID-19 patients are associated with brain injury, i.e. elevated Tau levels). The authors describe their hypothesis and gives good highlight about their results and about the further application of their results.

Introduction: gives a good summary about the scientific problem and what we already know about the phenomenon. The authors use clear wording, it is easy to follow and understand. It is also important to note that the question what Orsini and co-workers raised is important and needed to be solved.

Materials and methods: Materials and methods used in the study are written in a detailed way. It is easy to follow the experimental procedures. I have one very minor concern regarding to section 4.1. the source of HEK293 and N2a cells are not given. Plus, in section 4.2 it is not described if hiPSC-derived neurons were differentiated according to the manufacturer’s instructions or if it was based on a previously published paper.

Results: Results are shown in a good logical order. Figures are well integrated in the text part. In case of Figure 3A, it is not clear that what is the difference between the two WTs and the two JNPL3s. Please describe, because now it is confusing.

Figures: Results are organized in 5 figures. Image resolutions, especially in microscopy images are high enough to be legible. Figure legends support the understanding of the results represented.

Discussion: Another nicely organized and well written section of the manuscript. In the first part, we get a short summary about the results which are then taken into a wider scientific context. What is also a great advantage of the manuscript is that the authors highlight the potential limitation of the study (lack of using whole SARS-CoV-2 virus), however, the authors logically explain their decision which is absolutely acceptable.

English language: no issues were detected. The manuscript is free from typographical errors

After correcting the minor issues, the manuscript will be suitable for publication.
